# *WorldCompass*: Reinforcement Learning for Long-Horizon World Models

**Zehan Wang** [1,2] **Tengfei Wang** [2] † **Haiyu Zhang** [2] **Xuhui Zuo** [2] **Junta Wu** [2] **Haoyuan Wang** [2]
**Wenqiang Sun** [2] **Zhenwei Wang** [2] **Chenjie Cao** [2] **Hengshuang Zhao** [3] **Chunchao Guo** [2] † **Zhou Zhao** [1] ‡

## Abstract

This work presents *WorldCompass*, a novel Reinforcement Learning (RL) post-training framework for the long-horizon, interactive video-based world models, enabling them to explore the world more accurately and consistently based on interaction signals. To effectively "**steer**" the world model's exploration, we introduce three core innovations tailored to the autoregressive video generation paradigm: 1) *Clip-level rollout Strategy*: We generate and evaluate multiple samples at a single target clip, which significantly boosts rollout efficiency and provides fine-grained reward signals. 2) *Complementary Reward Functions*: We design reward functions for both interaction-following accuracy and visual quality, which provide direct supervision and effectively suppress reward-hacking behaviors. 3) *Efficient RL Algorithm*: We employ the negative-aware fine-tuning strategy coupled with various efficiency optimizations to efficiently and effectively enhance model capacity. Evaluations on the SoTA open-source world model, WorldPlay, demonstrate that WorldCompass significantly improves interaction accuracy and visual fidelity across various scenarios. Project page and online demo can be found: https://3d-models.hunyuan.tencent.com/world/

## 1. Introduction

World models enable humans or agents to interact with generative world environments. The emergence of Cosmos (Agarwal et al., 2025; Ali et al., 2025; Azzolini et al., 2025), HY-World (HunyuanWorld, 2025; Huang et al., 2025; Sun et al., 2025) and Genie series (Bruce et al., 2024; Parker-Holder et al., 2024; Ball et al., 2025), have significantly advanced the field, demonstrating the profound impact of video-based world models on both fundamental AI research

and practical generative media applications (Liao et al., 2025; Liu et al., 2025c; Ding et al., 2025; Li et al., 2025c; Liu et al., 2025d; Team et al., 2025a; Li et al., 2025b).

However, despite this immense potential, current open-source video-based world model solutions remain predominantly confined to the pre-training stage. These methods (Tang et al., 2025; He et al., 2025c; Sun et al., 2025; Yu et al., 2025) typically rely on pixel supervision from raw visual data to implicitly learn how to follow input actions.

In this work, we focus on post-training for long-horizon video-based world models. We attempt to employ reinforcement learning (RL) to more directly teach the model to explore the world with better accuracy and consistency, grounded in interaction signals. To achieve this, we propose *WorldCompass*, a novel RL framework specifically designed to "**steer**" the world exploration.

Specifically, we redesign each stages of RL process, grounding them in the autoregressive, interactive and long-horizon generation paradigm of world model. 1) We introduce clip-level rollout for autoregressive video generation. This strategy significantly boosts both rollout efficiency and the granularity of the reward signal. Besides, it compels the model to rely on its own imperfect predictions, thus effectively mitigating the challenge of exposure bias. 2) We design two complementary reward functions tailored to the main characteristics of world modeling: action following and visual quality of the generated content. This complementary reward feedback effectively suppresses reward hacking. 3) We utilize the negative-aware fine-tuning strategy for RL training, supplemented by a suite of efficiency optimizations. Together, these methods drive the world model toward the desired direction, ensuring both robustness and efficiency.

We evaluate the *WorldCompass* framework by performing post-training on WorldPlay (Sun et al., 2025), a recent state-of-the-art open-source world model. Through comprehensive evaluation, we find that our RL training substantially improves the model's interaction accuracy and visual quality across various scenarios: varying durations (from short-term to long-term), and different action complexities (basic actions or composite actions). This comprehensive improvement demonstrates that WorldCompass possesses high generalizability and effectively strengthens the model's fundamental capabilities.

---
[1]Zhejiang University [2]Tencent Hunyuan [3]The University of Hong Kong. Correspondence to: Chunchao Guo <chunchaoguo@gmail.com>, Zhou Zhao <zhaozhou@zju.edu.cn>.

*Proceedings of the 43$^{rd}$ International Conference on Machine Learning*, Seoul, South Korea. PMLR 306, 2026. Copyright 2026 by the author(s).

Our contribution can be summarized as follows:

- We highlight the value of post-training for advanced world models and introduce *WorldCompass*, a novel RL framework specifically designed for video-based world models.

- We dive into the autoregressive, interactive and long-horizon characteristics of the world model, redesigning the RL framework to achieve efficient training and fine-grained feedback signals.

- We provide a comprehensive evaluation on WorldPlay, demonstrating that RL post-training significantly enhances the capabilities of this state-of-the-art open-source world model across various scenarios.

## 2. Related Work

### 2.1. Video-based World Model

World model aims to predict future states, adhering to physical and geometric laws based on current and past observations and actions. It enables users or agents to interact with any environment. The recent Genie series (Bruce et al., 2024; Parker-Holder et al., 2024; Ball et al., 2025) demonstrate the significant potential of video-based world models in embodied intelligence and content creation. This approach utilizes a video generation model (Google, 2025; Wu et al., 2025; Wan et al., 2025) to interactively generate videos and explore the world by subjecting the generation process to a discrete action signal. This intrinsic need for interactive exploration requires the generation process to be autoregressive and long-horizon, and simultaneously demands precise fidelity to diverse action control conditions.

Recent work revolves around these requirements. Diffusion Forcing (Chen et al., 2024) enables the autoregressive generation of long video clips by using variable timesteps during training. Another line of works (Wang et al., 2024; He et al., 2024; Valevski et al., 2024) embed discrete or continuous control signals into the video generation model to govern camera movements within the generated video. More recent efforts (Yu et al., 2025; Li et al., 2025a; Sun et al., 2025; Fan et al., 2025) integrate both aspects, allowing models to autoregressively generate video clips following action conditions and finally compose long-horizon sequences.

However, these methods primarily focus on the pre-training stage, where models implicitly learn to follow actions through pixel supervision from videos. This approach limits the existing methods' ability over action switching or complex composite actions. In contrast, our approach introduces a post-training phase for the world model. By providing direct supervision for both action fidelity and visual quality, we significantly enhance the model's capabilities.

### 2.2. Reinforcement Learning

**RL for Autoregressive LLM** The recent success of DeepSeek-R1 (Guo et al., 2025) demonstrates that large-scale on-policy Reinforcement Learning (RL), when coupled with a reliable reward function, can guide autoregressive LLM towards emergent capabilities growth. The GRPO (Shao et al., 2024) algorithm utilized in DeepSeek-R1 has attracted significant attention. By leveraging the mean and variance of policy groups, GRPO removes the need for a separate value network (Schulman et al., 2017), resulting in a more memory-efficient approach. The effectiveness of on-policy RL has been wildly validated in large-scale experiments of LLMs (Yang et al., 2025; Liu et al., 2025a; Zheng et al., 2025a).

**RL for Diffusion Model** Inspired by the success of RL in LLMs, recent research has explored adapting RL algorithms for the post-training of diffusion models. While DiffusionDPO (Wallace et al., 2024) achieves alignment using off-policy preference pairs, more recent works like Flow-GRPO (Liu et al., 2025b) and Dance-GRPO (Xue et al., 2025) have successfully integrated the GRPO algorithm with diffusion models. By utilizing SDE solvers (Song et al., 2020) to enable on-policy RL, these methods have demonstrated significant performance gains. Furthermore, DiffusionNFT (Zheng et al., 2025b) builds upon the concept of group-wise advantage estimation, combining it with negative-aware fine-tuning to provide a more computationally efficient and effective refinement process.

However, current RL framework for diffusion models primarily target the paradigm where the entire content sequence (e.g., image, video, or audio) is generated in parallel within a single diffusion process. In contrast, world models necessitate sequential generation in an autoregressive manner, and often involve very long-horizon sequences. This fundamental architectural shift prevents the direct application of existing RL pipelines to our task. To bridge this gap, we propose a novel RL framework specifically tailored for the unique requirements of video-based world models.

## 3. *WorldCompass*

### 3.1. Preliminaries

**Interactive World Modeling** The core concept of world model is a generative system that enables a human or agent to interact with a world environment. Following the definition of Genie 2 (Parker-Holder et al., 2024), we frame this problem as autoregressive streaming video generation. This process is realized through a video diffusion model, denoted as $\pi_\theta(x_n|x_{1:n-1}, a_n, c)$, which generates the future state (i.e., the next video clip) $x_n$ conditioned on the history of world observations $x_{1:n-1}$, the user's interaction actions $a_n$, and a textual or visual world prompt $c$.

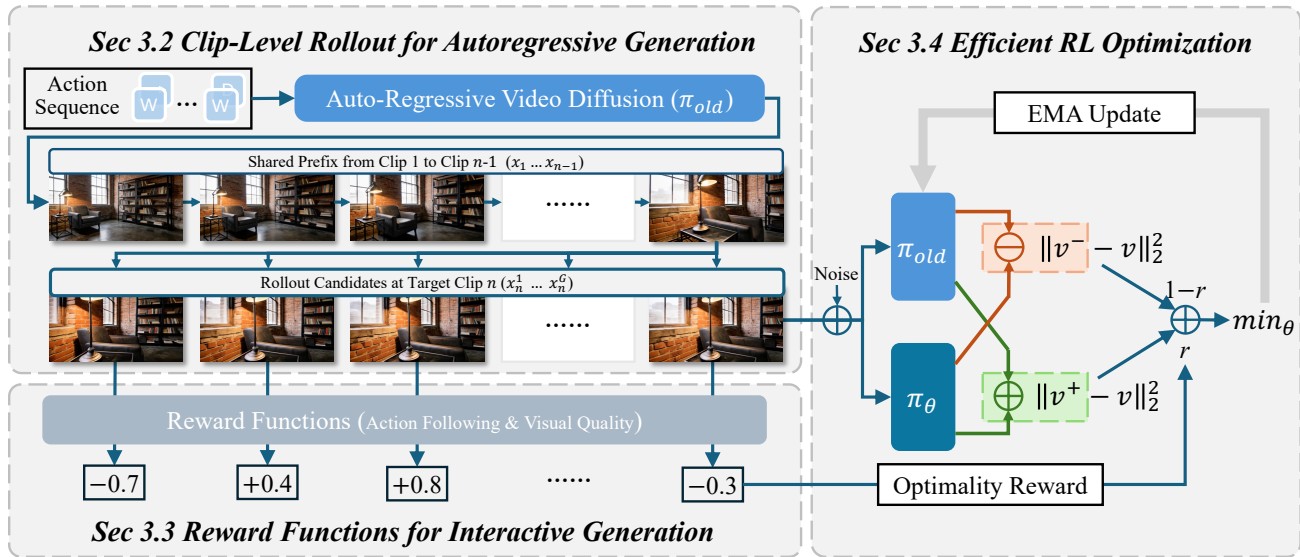

*Figure 1.* **Overview of *WorldCompass*.** 1) Starting from environmental prompts and action sequences, we generate shared prefix video clips. At the $n$-th target clip, we perform clip-level rollouts to generate a set of candidate video clips. 2) We design reliable reward functions to evaluate the action-following accuracy and visual quality of rollout samples. 3) We employ efficient RL algorithm to optimize the model, steering it toward generating high-scoring video clips.

**Reinforcement Learning for Diffusion Models** The fundamental loop of on-policy RL comprises three core stages: 1) *Rollout*: a group of $G$ rollouts is generated from a pre-trained diffusion model; 2) *Evaluation*: each sample is scored according to a set of predefined reward functions; and 3) *Optimization*: these generated samples and their associated rewards are utilized to update the model, incentivizing the generation of high-reward trajectories.

To employ RL to autoregressive, interactive and long-horizon video-based world models, we should address three corresponding challenges:

1. How to generate rollout samples for the autoregressive video generation model? (Sec. 3.2)

2. How to design reward functions that reliably evaluate interactive generation? (Sec. 3.3)

3. How to effectively and efficiently optimize world modeling using RL? (Sec. 3.4)

### 3.2. Clip-level Rollout for Autoregressive Generation

As the most typical autoregressive generation model, we first study how a LLM performs its rollout. Given an input prompt $c$, the LLM model $\pi_{\text{llm}}$, and the reward function $r$, RL scheme for LLMs would independently generate $G$ sequences and score each. The computation process for the $i$-th sample $x_{1:n}^{(i)}$ and its corresponding score $s^{(i)}$ can be formulated as:

$$x_{1:n}^{(i)} = \pi_{\text{llm}}(\cdot|c), \; s^{(i)} = r(x_{1:n}^{(i)}, c) \quad (1)$$

However, we find that this intuitive sequence-level rollout scheme is fundamentally unsuitable for autoregressive video generation. The core issue is that the feedback signal is too sparse. A single, final reward score for an entire video sequence cannot precisely identify which individual clips adhere to their action conditions or pinpoint when visual quality degradation begins.

To address this, we formulate clip-level rollout strategy specifically for autoregressive video generation. Considering semantic textual or image condition $c$, a sequence of action conditions $a_{1:N}$ and a target clip index $n$ to rollout. we first autoregressively generate the preceding $n-1$ clip. Subsequently, we generate $G$ candidate rollouts for the $n$-th target clip. The generation and reward evaluation of the $i$-th sample at the $n$-th clip are formulated as follows:

$$x_{1:n-1} = \pi_\theta(\cdot|a_{1:n-1}, c) \quad (2)$$

$$x_n^{(i)} = \pi_\theta(\cdot|x_{1:n-1}, a_n, c), s^{(i)} = r(x_n^{(i)}, a_n, c) \quad (3)$$

The clip-level rollout strategy offers two key advantages:

1. **Rollout Efficiency:** For world modeling, long-term interactive generation is crucial. Meanwhile, for RL, a large number of rollout samples is vital for stable training and strong performance, implying that both the video length ($N$) and rollout number ($G$) should be large. With our clip-level rollout, the preceding $n-1$ clips are sampled only once and reused repeatedly. The model only performs $G$ repetitive samplings for the $n$-th clip. Consequently, the computational complexity of the rollout process is roughly reduced from $O(N \cdot G)$ to $O(N + G)$, which significantly improves rollout sampling efficiency.

2. **Consistent and Fine-grained Reward:** Generating rollout samples from identical historical observations mitigates the potential inconsistencies that diverse prefixes introduce into current predictions. Furthermore, by evaluating these samples within the same context, we derive a more granular and comparable reward signal. These advantages enable a more targeted optimization for the model to synthesize high-fidelity clips, ensuring that gradient updates are driven by the quality of the current generation rather than historical variance.

## 3.3. Reward Functions for Interactive Generation

The effectiveness of an RL framework depends on its reward functions, as they dictate the direction of model improvement. In the context of interactive video generation, we focus on rewarding two fundamental attributes: interaction following and visual quality. Accordingly, we implement specific reward functions to assess each sampled clip.

**Interaction Following Score** For world model, interaction following evaluates whether the generated clip moves as the given action condition indicated. Following recent work like Genie 3 (Ball et al., 2025), the action signal is decomposed into: translation and rotation.

To assess these actions, we utilize advanced 3D foundation model (Liu et al., 2025d; Lin et al., 2025), to estimate the camera trajectory within the generated clip. This continuous trajectory is then mapped to a predefined discrete action space to calculate adherence accuracy.

For rotation, we determine the action between adjacent frames by comparing the estimated relative camera rotation against a predefined threshold, $\tau_{\text{rot}}$. Evaluating translation action is more challenging because the detected positional scale varies across different scenes. Since a single universal threshold, $\tau_{\text{trans}}$, cannot generalize across all environments, we set multiple translation thresholds. A translation action is deemed correct if it matches the conditional input under any of the defined thresholds, ensuring a robust evaluation across diverse scenes.

Finally, to provide more discriminative guidance during optimization, we calculate the rotation and translation accuracies independently and define the final interaction following score as their average.

**Visual Quality Score** To evaluate the visual fidelity of the generated video, we employ HPSv3 (Ma et al., 2025) as our reward model, which inherently assesses both text-visual alignment and aesthetic quality. Specifically, we sample frames from each clip at 4-frame intervals and compute the average HPSv3 score across these frames to represent the overall visual quality.

**Discussion** Reward hacking is a common challenge in RL for generative models. Due to the inherent error accumulation characteristic in autoregressive video generation, world models are even more prone to such problem. In practice, we find that our two reward functions act as mutual regularizers. By balancing these two objectives, the framework prevents the model from optimizing one at the expense of the other, thereby suppressing reward hacking and leading to more robust training.

## 3.4. Efficient RL Optimization

**RL Algorithem** Preliminary post-training methods for world models (Ye et al., 2025; He et al., 2025a) typically rely on FlowGRPO (Liu et al., 2025b), which employs SDE sampling to achieve stochastic exploration by generating multiple samples from the same initial noise. However, our investigation reveals that SDE sampling over same noise only diversifies the visual scenes but leaves the camera movement virtually unchanged. This limitation in exploring diverse camera trajectories constrains the potential improvement of the action following capabilities.

Inspired by DiffusionNFT (Zheng et al., 2025b), we perform policy optimization using a negative-aware fine-tuning strategy. The rollout data are sampled from different initial noises, and the model is directly trained with the flow matching objective. Given $G$ samples at the $n$-th video clip index, $\{x_n^{(i)}\}_{i=1}^G$, with corresponding interaction following scores $\{s_{\text{IF}}^{(i)}\}_{i=1}^G$ and visual quality scores $\{s_{\text{VQ}}^{(i)}\}_{i=1}^G$, we first compute the advantage for each reward dimension:

$$a_j^{(i)} = \frac{s_j^{(i)} - \text{mean}(\{s_j^{(i)}\}_{i=1}^G)}{\text{std}(\{s_j^{(i)}\}_{i=1}^G)}, \quad \text{for } j \in \{\text{IF, VQ}\} \quad (4)$$

Then, we derive the optimality probability $r^{(i)}$ of the $i$-th sample through a clipped linear combination of the two normalized advantages:

$$r^{(i)} = \frac{1}{2} + \frac{1}{2} \text{clip}\left[\frac{\lambda a_{\text{IF}}^{(i)} + (1-\lambda)a_{\text{VQ}}^{(i)}}{Z}, -1, 1\right] \quad (5)$$

where $\lambda$ is the trade-off hyperparameter, and $Z$ is the normalizing factor. The final optimization loss is defined as:

$$\mathcal{L}(\theta) = \mathbb{E}_{\substack{t \sim T \\ i \sim G \\ n \sim N}}\left[r^{(i)}\left\|v_\theta^+ - v^{(i)}\right\|_2^2 + (1-r^{(i)})\left\|v_\theta^- - v^{(i)}\right\|_2^2\right] \quad (6)$$

$where$
$$z_t^{(i)} = (1-t)x_n^{(i)} + t\epsilon; \ \epsilon \sim \mathcal{N}(0, I)$$
$$v_\theta^+ = (1-\beta)v_{\theta_{\text{old}}}(z_t^{(i)}, x_{1:n-1}, a, c, t) + \beta v_\theta(z_t^{(i)}, x_{1:n-1}, a, c, t)$$
$$v_\theta^- = (1+\beta)v_{\theta_{\text{old}}}(z_t^{(i)}, x_{1:n-1}, a, c, t) - \beta v_\theta(z_t^{(i)}, x_{1:n-1}, a, c, t)$$
$$v^{(i)} = x_0^{(i)} - \epsilon$$

---

**Algorithm 1** *WorldCompass* Training Process

---

**Require:** Initial policy model $\pi_\theta$, its EMA copy $\pi_{\theta_{\text{old}}}$; reward functions $R_{\text{IF}}, R_{\text{VQ}}$; prompt and action dataset $\mathcal{D}$
**Ensure:** Optimized policy model $\pi_\theta$

 1: **for** each training iteration $k$ **do**
 2:     Sample batch $\mathcal{D}_b \sim \mathcal{D}$        *// Clip-Level Rollout & Rewards, Sec. 3.2 & 3.3*
 3:     Select target clip index for progressive training: $n = (k \bmod N) + 1$
 4:     **for** each $(c, a_{1:n}) \in \mathcal{D}_b$ **do**
 5:        Generate shared prefix $x_{1:n-1}$ and $G$ rollout clips $\{x_n^{(i)}\}_{i=1}^G$ using $\pi_{\theta_{\text{old}}}$
 6:        Compute rollout samples' advantages $\{a_{\text{IF}}^{(i)}, a_{\text{VQ}}^{(i)}\}_{i=1}^G$ with reward functions
 7:        Compute optimality probability $r^{(i)}$ by combining and clipping the advantages
 8:     **end for**
 9:     Subsample Best-of-N samples $\mathcal{G}_{\text{sub}} \subset 1:G$        *// Efficient RL Optimization, Sec. 3.4*
10:     Subsample a random set of timesteps $\mathcal{T}_{\text{sub}} \subset 1:T$
11:     **for** $i \in \mathcal{G}_{\text{sub}}$ and $t \in \mathcal{T}_{\text{sub}}$ **do**
12:        Forward diffusion process: $z_t^{(i)} = (1-t)x_n^{(i)} + t\epsilon$; $v^{(i)} = x_n^{(i)} - \epsilon$
13:        Calculate implicit positive/negative velocity $v_\theta^+, v_\theta^-$ (Eq. 6)
14:        Compute weighted loss: $\mathcal{L}_i = r^{(i)} \left\| v_\theta^+ - v^{(i)} \right\|_2^2 + (1 - r^{(i)}) \left\| v_\theta^- - v^{(i)} \right\|_2^2$
15:        Update policy: $\theta \leftarrow \theta - \lambda_{\text{lr}} \nabla_\theta \mathcal{L}_i$
16:     **end for**
17:     Update old policy: $\theta_{\text{old}} \leftarrow \eta\,\theta_{\text{old}} + (1-\eta)\theta$
18: **end for**

---

where the $T$ denotes the diffusion sampling timesteps, and $N$ represents the maximum length of clips used in training. The $v_{\theta_{\text{old}}}$ and $v_\theta$ are predictions from the old model for rollout sampling and the current model undergoing training. Both models are initialized from the base model, and the old model is EMA updated by the trained model.

Notably, we omit the KL divergence loss used in the original DiffusionNFT (Zheng et al., 2025b). While KL regularization between the training and original model's outputs typically mitigates mode collapse and reward hacking, we empirically observe limited performance improvement when including it. Instead, we employ a lower learning rate and the EMA update strategy to prevent over-optimization and achieve superior final results.

**Efficient Training Strategy**  As shown in Eq. 6, the optimization theoretically involves iterations over the diffusion sampling timesteps ($T$), the number of rollout samples ($G$), and the maximum length of clips ($N$). This process is time-consuming and computationally intensive. In practice, we employ several strategies to accelerate the training process:

1. For sampling timestep $T$, following observations in previous work (Xue et al., 2025; Liu et al., 2025b), we randomly select subsets of timesteps within a denoising trajectory for each training iteration, rather than processing all $T$ steps. This significantly reduces computational overhead without compromising performance.

2. For rollout samples $G$, we adopt the Best-of-N selection strategy. Specifically, we select only the subsets of samples

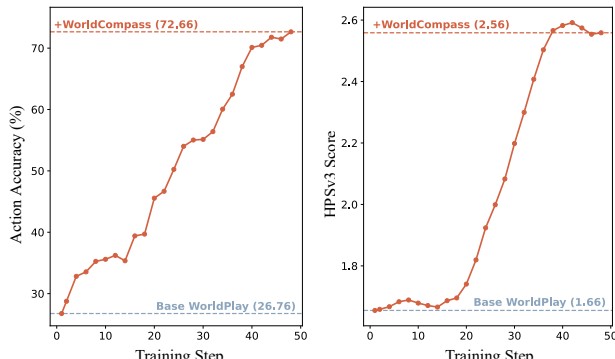

*Figure 2.* Evolution of interaction following and visual quality scores during the RL training of WorldPlay (HunyuanVideo-1.5 version). These reward metrics are evaluated on a fixed subset of the test set with complex combined action.

corresponding to the top 3 and bottom 3 rewards for training. This approach enhances training efficacy by focusing the model on the most informative cases, leveraging high-reward samples for reinforcement and low-reward samples for error correction.

3. For clips length $N$, we implement a progressive optimization strategy. During training, the target clip index $n$ (used for rollout and optimization) cycles incrementally from 1 to $N$, as shown in Line 3 of Alg. 1. This scheduling naturally induces a curriculum learning effect, adhering to RL principles by gradually increasing the task horizon. Furthermore, maintaining a unified video length across all parallel compute nodes maximizes hardware utilization and sampling efficiency.

# 4. Experiments

## 4.1. Experimental Setup

**Base Model** We evaluate our framework using the recent WorldPlay (Sun et al., 2025) as our base world model. For a rigorous validation, we test two distinct variants of World-Play: HunyuanVideo-1.5-8B (Wu et al., 2025) and Wan2.2-5B (Wan et al., 2025). These models utilize eight basic actions: forward, backward, left, right movement, and up, down, left, right rotation. These individual basic movements can also be combined into complex compositional actions. Under the autoregressive generation paradigm, the video token sequence is processed as discrete chunks, where each chunk corresponds to a 16-frame video clip. To optimize the model for long-horizon generation, we set the maximum generation length $N$ to 16 clips, totaling approximately 256 decoded frames.

**Training Data** Since all supervision is derived directly from reward functions, our framework eliminates the need for explicit manual annotations. We utilize a training set representative of real-world inference inputs, consisting of 4,000 diverse images and corresponding caption. Furthermore, to enhance the model's robustness in challenging scenarios, we randomly constructed complex action sequence, primarily consisting of combinations of the basic actions.

**Hyper-parameter** During training, each step comprised 64 groups with a rollout group size of $G = 16$. The sampling step $T$ for rollout is set to 40, from which 50% of the sub-sampling steps are randomly selected for training. The rotation threshold for the action-following score is set to $1°$. Based on the inherent predicted position scale of 3D foundation models, the multiple translation thresholds are set to $[0.01, 0.02, 0.03, 0.04, 0.05]$. The parameter $\lambda$ and normalizing factor $Z$ in Eq. 5 is 2/3 and 2 respectively. The parameter $\beta$ in Eq. 6 is 1. We use the Muon (Team et al., 2025b) optimizer with a learning rate of 1e-5, and the EMA factor for the old model is linearly annealed from 0.4 to 0.8. The training process spans 3 days across 64 H20 GPUs.

## 4.2. Main Results

**Evaluation Protocol** We evaluate our model using 600 cases from the WorldPlay test set. To better simulate real-world application scenarios, we redesign the action control sequences for each test sample into two categories: basic actions and composite actions. To comprehensively assess the model's capabilities across various conditions, we evaluate different model variants across three video lengths: short ($\sim 125$ frames), medium ($\sim 253$ frames), and long ($\sim 381$ frames). We report the average action-following accuracy and the HPSv3 visual quality score, both computed across all generated clips using a 4-frame sampling interval.

| | | Combined Action | | Basic Action | |
|---|---|---|---|---|---|
| | | Acc$_{action}$ | HPSv3 | Acc$_{action}$ | HPSv3 |
| Short-term (125 frames) | HY-Video-1.5 | 21.74 | -1.05 | 62.33 | 1.96 |
| | + *WorldCompass* | 58.20 | 0.42 | 68.62 | 3.77 |
| | Δ | +36.46 | +1.47 | +6.29 | +1.81 |
| | Wan2.2 | 22.87 | -1.10 | 58.28 | 1.83 |
| | +*WorldCompass* | 49.81 | 0.20 | 64.72 | 3.17 |
| | Δ | +26.94 | +1.30 | +6.44 | +1.34 |
| Mid-term (253 frames) | HY-Video-1.5 | 19.73 | -0.19 | 63.35 | 1.91 |
| | + *WorldCompass* | 55.01 | 0.37 | 74.09 | 3.61 |
| | Δ | +35.28 | +0.56 | +10.74 | +1.70 |
| | Wan2.2 | 20.33 | -1.67 | 57.94 | 1.63 |
| | + *WorldCompass* | 50.32 | 0.27 | 63.87 | 3.37 |
| | Δ | +29.99 | +1.94 | +5.93 | +1.74 |
| Long-term (381 frames) | HY-Video-1.5 | 19.70 | -0.33 | 64.28 | 1.90 |
| | + *WorldCompass* | 54.82 | 0.73 | 76.56 | 3.72 |
| | Δ | +35.12 | +1.06 | +12.28 | +1.82 |
| | Wan2.2 | 19.58 | -0.80 | 55.59 | 1.91 |
| | + *WorldCompass* | 42.92 | 0.59 | 63.91 | 3.59 |
| | Δ | +23.34 | +1.39 | +8.32 | +1.68 |

*Table 1.* Quantitative Results on two version of WorldPlay: HunyuanVideo-1.5-8B (HY-Video-1.5) and Wan2.2-5B (Wan-2.2). We compare the performance of two model variants across different scenarios: 1) the base model and 2) the model after WorldCompass post-training. The better results are highlighted in **bold**.

**Quantitative Results** Fig. 2 illustrates the evolution of reward scores throughout the RL training process. Notably, our method achieves significant performance gains in both interaction following and visual quality on challenging action inputs within a remarkably small number of training steps. In Table 1, we provide a comprehensive comparative analysis. Overall, WorldCompass yields substantial performance improvements across various base model versions and different video lengths. Specifically, for complex composite action inputs, our method improves average accuracy from approximately 20% to 55%. For basic action inputs, we observe a 10% improvement in interaction accuracy. Beyond achieving superior interaction following, the RL training also enhances the overall visual quality of the generated videos.

It is worth noting that our action accuracy is computed by matching every 4 frame with its corresponding action condition, which represents a rigorous evaluation standard. At lower score ranges such as 10% to 30%, errors primarily stem from the failure of the model to comprehend and execute the input action. At higher ranges between 50% and 60%, errors are mainly characterized by latencies during action switching. Consequently, the jump from 20% to 55% for complex actions represents a fundamental shift from failing to follow actions to successfully executing them. For simple actions, the improvement from 60% to 70% reflects more rapid responses during action switching.

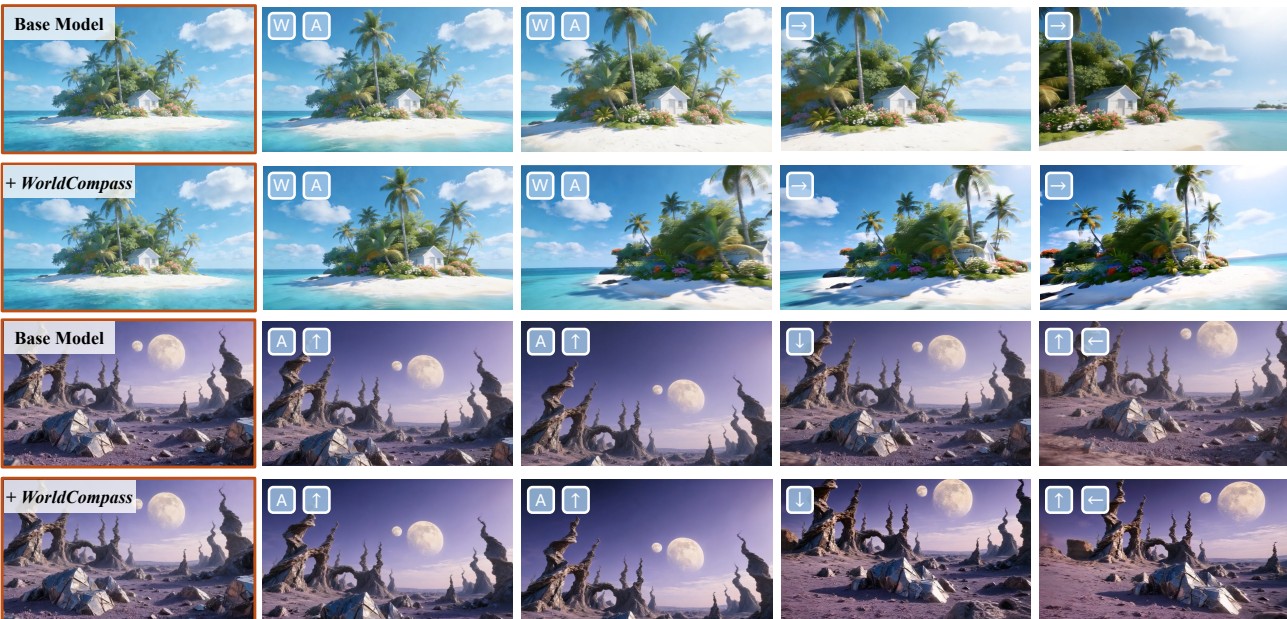

Figure 3. Qualitative comparisons under complex combined action sequence.

| Row | Rollout Type | IF Score | VQ Score | RL Algorithm | Combined Action | | Basic Action | |
|---|---|---|---|---|---|---|---|---|
| | | | | | Acccaction | HPSv3 | Acccaction | HPSv3 |
| 0 | - | - | - | - | 19.70 | -0.33 | 64.28 | 1.90 |
| 1 | clip-level | ✓ | ✓ | DiffusionNFT | 54.82 | 0.73 | 76.56 | 3.72 |
| 2 | sample-level | ✓ | ✓ | DiffusionNFT | 12.45 | 0.19 | 58.42 | 2.69 |
| 3 | clip-level | ✓ | ✗ | DiffusionNFT | 36.39 | -2.67 | 67.60 | -1.83 |
| 4 | clip-level | ✗ | ✓ | DiffusionNFT | 11.51 | 1.01 | 35.94 | 4.19 |
| 5 | clip-level | ✓ | ✓ | DanceGRPO | 20.02 | 0.59 | 67.43 | 3.97 |

Table 2. Ablation study for the core components in WorldCompass. Line 0 provide the results of baseline without RL training.

**Qualitative Results** In Fig. 3 and 4, we provide visual comparisons of generated results with and without RL for both complex combined actions and simple basic action sequences, respectively. The qualitative results align with our quantitative findings: the WorldCompass post training framework significantly enhances the ability of the model to follow interactions while improving visual quality. Furthermore, as discussed in (He et al., 2025b), 3D foundation model successfully predict camera trajectories from a generated video clip that match the input condition also indirectly implies that the content possesses spatial geometric consistency. We also observe that the generated content demonstrates improved spatial consistency.

### 4.3. More Discussion

To further dissect our framework, we conduct more in-depth ablation studies. All the experiments are performed on the HY-Video-1.5 version of WorldPlay, and all results are reported under long-term generation setting.

| Subset of Timestep | Best-of-N Sampling | Acc$_{action}$ | HPSv3 | Iteration Time |
|---|---|---|---|---|
| ✓ | ✓ | 54.82 | 0.73 | 1.00× |
| ✓ | ✗ | 55.28 | 0.75 | 1.42× |
| ✗ | ✗ | 54.68 | 0.78 | 2.26× |

Table 3. Ablation study of efficiency optimization strategies. The results are reported under combined action setting.

**Sample- or Clip-level Rollout** Comparing rows 0,1,2 in Tab. 2, we can summarize that a clip-level rollout is crucial for the effectiveness of WorldCompass. Specifically, sample-level rollout tends to degrade the action following capability of the model, yielding only minor gains in visual quality. This phenomenon occurs because sample-level rollout results in an overly sparse reward density for long-duration videos. While visual quality often exhibits strong temporal dependencies that allow a holistic score to represent overall visual fidelity, action-following accuracy remains relatively independent across clips. Consequently, an aggregate sample-level score for action following averages out

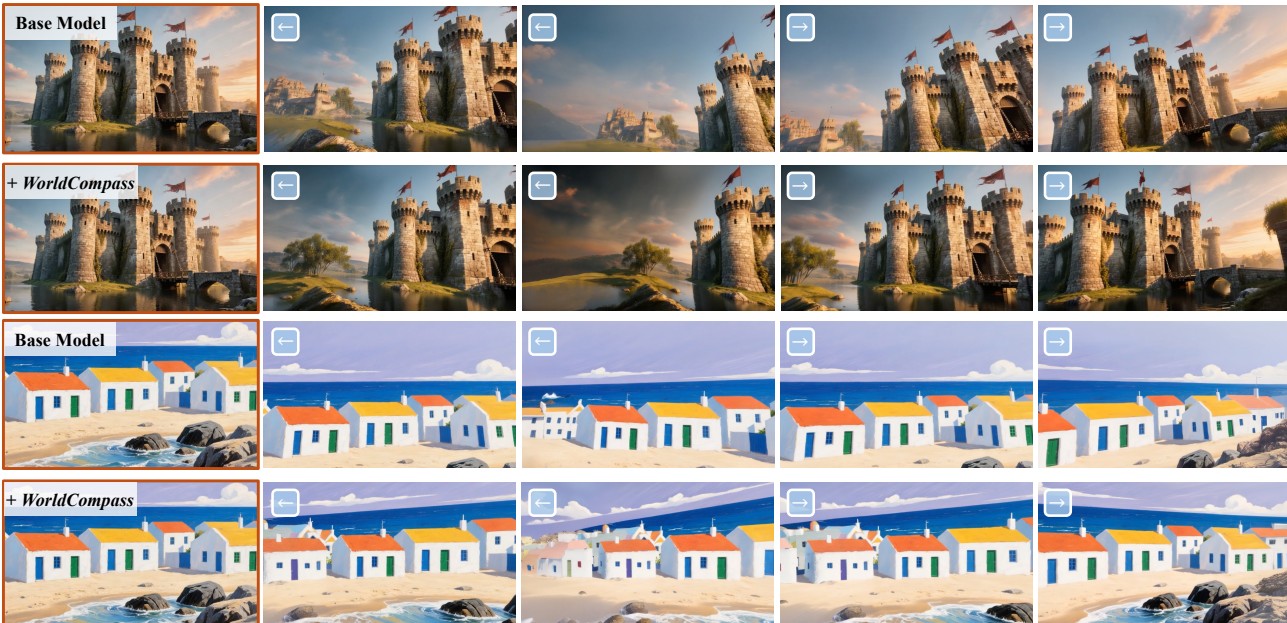

*Figure 4.* Qualitative comparisons under simple basic action sequence.

specific successes and failures, thereby failing to provide a discriminative reward signal. This coarse grained signal becomes uninformative or even misleading, ultimately hindering the model from improving its core capabilities.

**Complementary Reward Functions**  We study the effect of individual reward functions by comparing rows 1, 3, and 4 in Tab. 2. Our results indicate that relying on a single reward function makes the model highly susceptible to reward hacking. Specifically, optimizing solely for the Interaction Following (IF) Score improves action accuracy but triggers a noticeable decline in visual quality. This degradation in visual fidelity severely undermines training stability, often leading to model collapse. Conversely, using only the Visual Quality (VQ) Score yields aesthetically pleasing results but produces static or motionless content. When both rewards are applied simultaneously, they act as a mutual constraint. This synergy guides the model toward the desired direction of improvement, achieving superior performance across both dimension.

**Alternative RL Algorithm**  In row 4 of Tab. 2, we valuate the necessity of DiffusionNFT (Zheng et al., 2025b) by replacing it with DanceGRPO (Xue et al., 2025), another widely-used RL algorithm for diffusion models. As analyzed in Sec. 3.4, DanceGRPO generates rollout video clips with minimal camera motion variance. This lack of diversity restricts the search space for optimal policies within the world modeling task, which ultimately results in marginal improvements in interaction-following capability.

**Training Efficiency**  To investigate the impact of our efficiency optimization strategies on final performance and training iteration time, we provide a detailed comparative study in Tab. 2. The results demonstrate that our strategies reduces training overhead by 50% while maintaining competitive results. Selecting a subset of timesteps and employing a Best-of-N sample selection strategy preserves the most critical information within the RL training process, thereby significantly enhancing overall efficiency.

## 5. Conclusion

We introduce *WorldCompass*, a novel online reinforcement learning framework tailored specifically for world models. Recognizing the interactive, long-horizon, and autoregressive nature of world modeling tasks, we redesign the RL training pipeline for diffusion models to better address these unique characteristics. Specifically, we introduce a clip-level rollout strategy that provides fine-grained rewards while boosting rollout efficiency for long-term autoregressive video generation. Furthermore, we incorporate complementary reward functions that act as mutual constraints to stably improve both interactive accuracy and visual quality. Finally, we utilize a negative-aware fine-tuning algorithm for RL model optimization. Experimental results demonstrate that WorldCompass significantly enhances the state-of-the-art world model, WorldPlay, particularly when handling challenging compositional action sequences.

## Impact Statements

This paper presents work whose goal is to advance the field of machine learning. There are many potential societal consequences of our work, none of which we feel must be specifically highlighted here.

## Acknowledgments

This work was supported by National Natural Science Foundation of China under Grant No.624B2128, and Project No.U24A20326.

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

## A. Limitation

Currently, there is a lack of reliable metrics to evaluate visual quality drift and spatial memory retention in long-form video generation. Consequently, our reward signal lacks the direct constraints to penalize such drift, which leads to cumulative quality degradation when applying large-scale RL training to long-duration autoregressive video generation. While we currently mitigate this issue through a conservative training strategy by employing fewer iterations and a reduced learning rate, a more fundamental solution would involve a robust reward function specifically designed to evaluate visual drift and spatial memory. This represents a promising direction for future research, although it remains beyond the scope of this work.

## B. More Qualitative Results

We provide additional qualitative results comparing the results before and after WorldCompass RL training in Fig. 5, 6, 7 and 8. To facilitate better visualization, we reconstructed the 3D scenes and camera trajectories from the generated videos. For a more intuitive comparison, we evaluate the models across multiple distinct scenarios using a consistent action sequence: the first half consists of the "W+A" command (moving forward-left), followed by a "→" command (turning right) in the second half. As illustrated, after undergoing WorldCompass training, the world model demonstrates significantly improved action-following accuracy and superior geometric consistency.

Base Model                                                    + WorldCompass

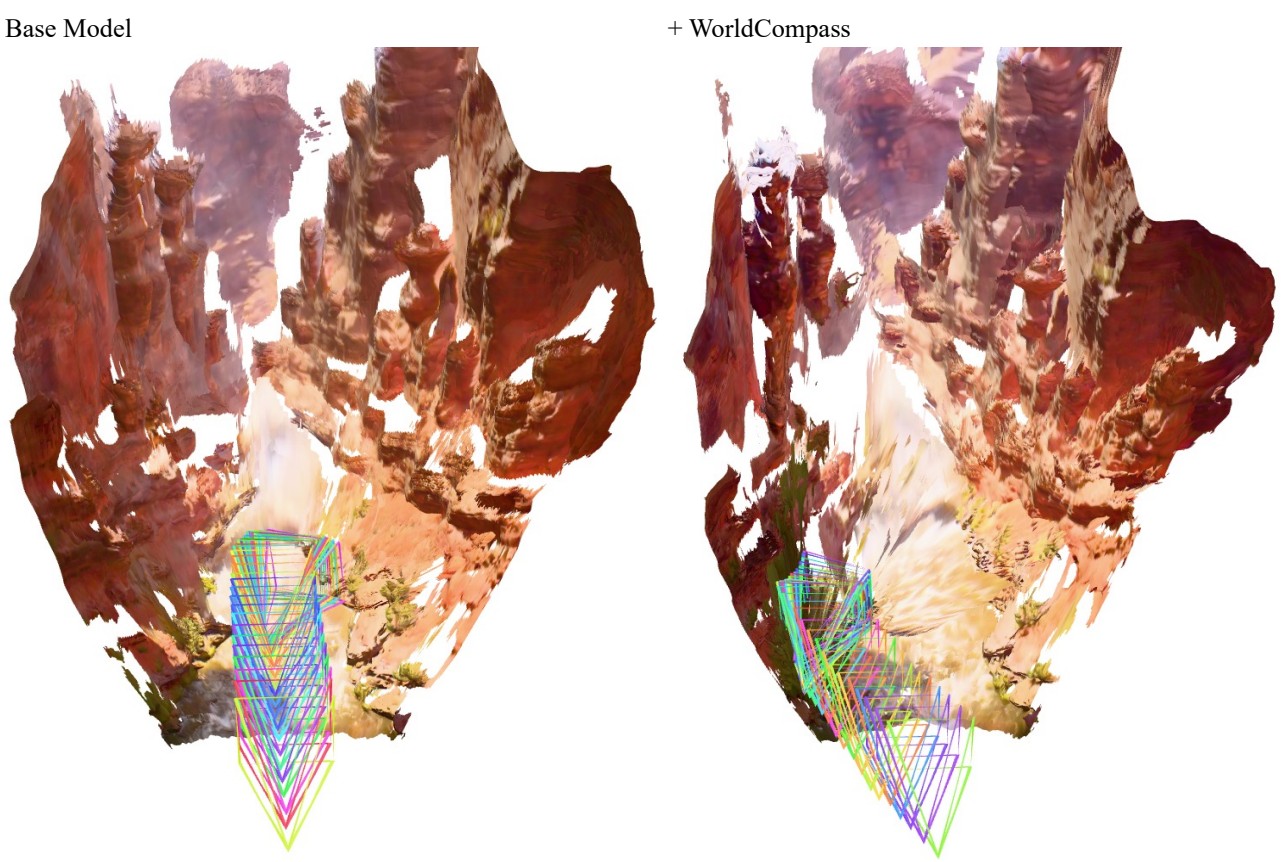

*Figure 5.* Visualization Case 1. The input action sequence consists of W+A" (moving forward-left) for the first half, followed by →" (turning right) in the second half.

Base Model

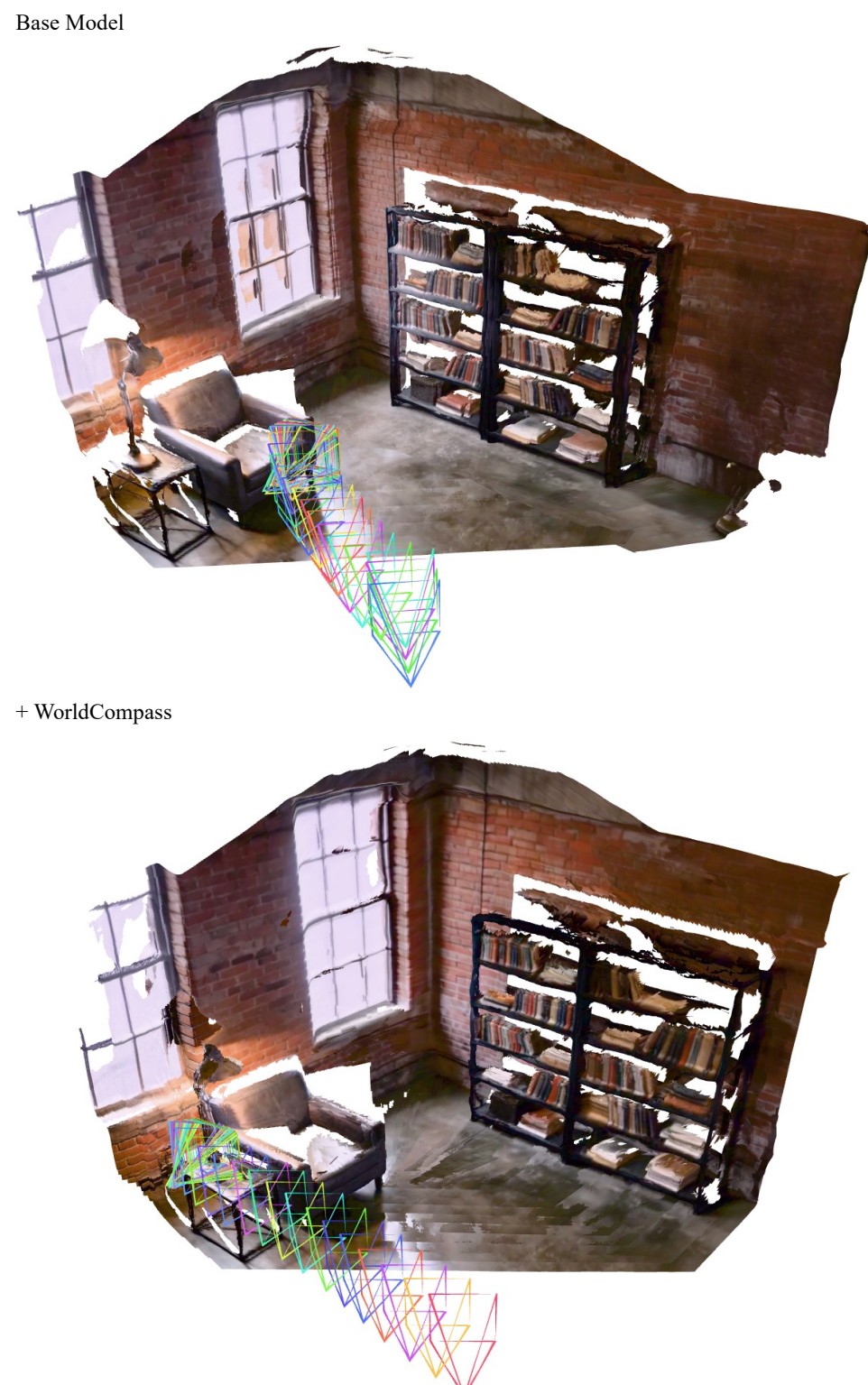

+ WorldCompass

*Figure 6.* Visualization Case 2. The input action sequence consists of W+A" (moving forward-left) for the first half, followed by →" (turning right) in the second half.

Base Model

+ WorldCompass

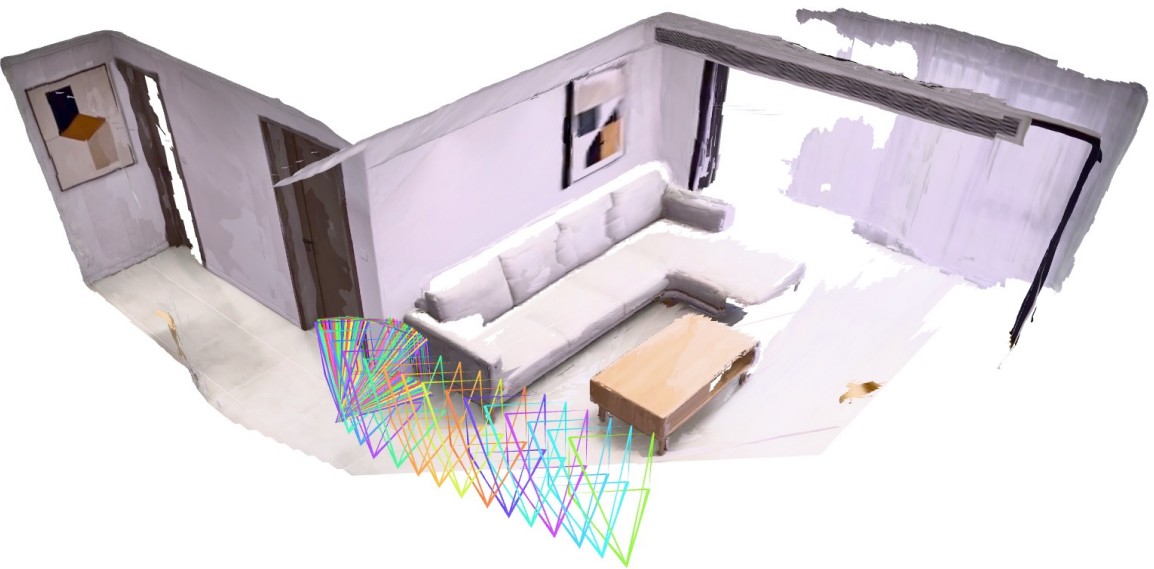

*Figure 7.* Visualization Case 3. The input action sequence consists of W+A" (moving forward-left) for the first half, followed by →" (turning right) in the second half.

Base Model

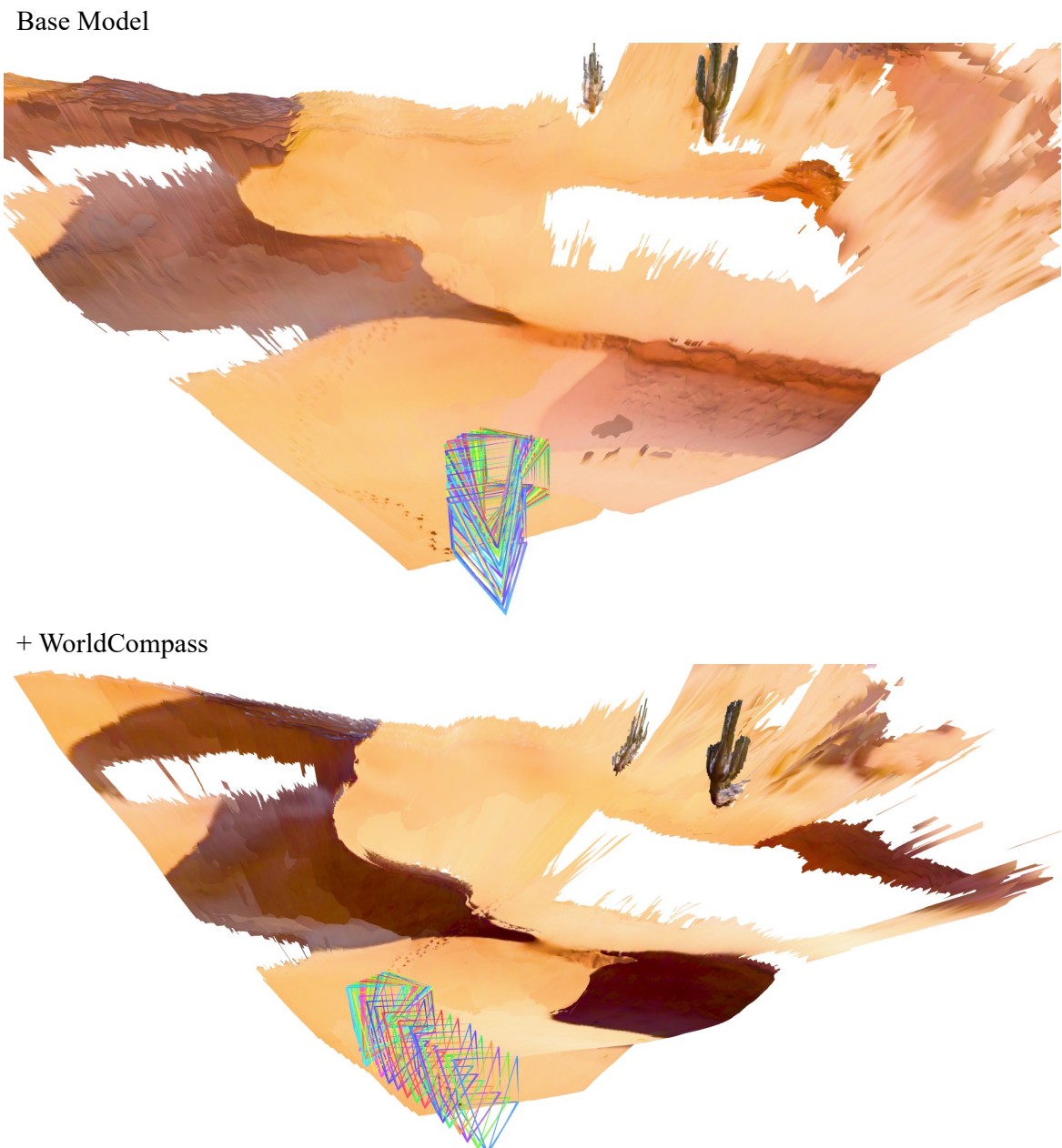

+ WorldCompass

*Figure 8.* Visualization Case 4. The input action sequence consists of W+A" (moving forward-left) for the first half, followed by →" (turning right) in the second half.

