# OpenReview forum: "WorldCompass: Reinforcement Learning for Long-Horizon World Models"
_ICML.cc/2026/Conference — ICML 2026 regular_

### Official Review · Reviewer_eCWH · 2026-02-28

**Soundness:** 2
**Presentation:** 3
**Significance:** 2
**Originality:** 2
**Overall Recommendation:** 3
**Confidence:** 3

**Summary:**

This work proposes *WorldCompass*, a tailored RL post-training framework for long-horizon video-based world models. Specifically, it dives into the detailed characteristics of such models and designs three techniques: 1) clip-level rollout for efficient rollout and fine-grained rewards; 2) complementary reward functions considering both action-accuracy and visual quality; 3) efficient RL algorithm using a negative-aware fine-tuning strategy.

**Compliance With Llm Reviewing Policy:**

Affirmed.

**Final Justification:**

My main concern is still about the limited type of evaluation tasks, i.e., only "first-person view." Plus, given the existence of RLVR, the concept of "training WM with RL" is not that novel, making the novelty of this paper kind of weakened. Overall, the novelty and the technical contribution of this work are not good enough, and therefore I'm inclined to maintain the weak reject.

**Key Questions For Authors:**

1. Could the evaluation consider metrics other than HPSv3 for assessing generation quality? Since HPSv3 is also used as the reward during training, the model may be overfitting to this metric. Using additional metrics could better demonstrate the actual improvement from RL.
2. Has the model been evaluated for consistency? For example, if the model executes “turn left” and then “turn right” to return to the original view, are the objects still consistent in the scene?

**Limitations:**

Yes

**Strengths And Weaknesses:**

### Strengths

1. The paper provides a thorough discussion of technical details and offers a detailed analysis of the challenges in designing RL algorithms for long-horizon world models.
2. The experiments include detailed ablations and analyses of different design choices, helping readers better understand the contribution of each component.

### Weaknesses

1. I believe the paper misses discussion of some key related work. For example, [1] also trains a world model using reinforcement learning. Although the technical details differ, the research theme is highly similar and should be acknowledged.
2. The proposed method seems overly tailored to “first-person view world models,” which focus only on changes in perspective and do not model other interactions with objects. In many scenarios, such as embodied AI, physical interactions with objects are also critical, but this paper appears to completely overlook such cases.

[1] Wu et al., "RLVR-World: Training World Models with Reinforcement Learning".

---

> ### Author Rebuttal · Authors · 2026-03-31
>
> ---
> **W1 Missed related work:**
>
> We appreciate the reviewer's suggestion. We will include a citation and a discussion of RLVR-World in the revised version of our manuscript.
>
> ---
> **W2 Overly tailored to “first-person view world models”:**
>
> We clarify that current limitations regarding interaction forms are primarily constrained by the underlying base world model. Existing solutions such as WorldPlay [1], Matrix-Game 2.0 [2], and LingBot-World [3], as well as the cases demonstrated in Google Genie 3 [4], are largely restricted to to first-person or third-person camera movements. Since the base model lacks the inherent capacity for richer interactions like object manipulation, it is not feasible to further enhance these specific skills during the RL post-training stage.
>
> Moreover, when a base model supports new control capabilities, we can strengthen the interaction-following performance by expanding the reward functions. For instance, to support object manipulation, we could integrate object detection [5], depth estimation [6], and orientation estimation [7] to evaluate the alignment between the generated object pose and the input manipulation condition.
>
> ---
> **Q1 Other generation quality metrics:**
>
> We provide additional evaluation results across a broader range of metrics, including LAION Aesthetic Score [8] and PickScore [9]:
>
> | Model           | Aesthetic | PickScore  |
> |-----------------|-----------|------------|
> | HY-Video-1.5    | 5.0762    | 19.2070    |
> | +WorldCompass   | 5.4344    | 19.3831    |
> | Δ               | +0.3583   | +0.1761    |
>
>
> ---
> **Q2 Spatial consistency:**
>
> To evaluate spatial consistency, we execute "turn left" followed by a "turn right" action across 100 test cases. Manual verification demonstrates that after WorldCompass post-training, the model achieves a 72% win rate in spatial consistency compared to the original base model. We attribute this improvement to the fact that interaction-following implicitly requires the model to maintain a coherent understanding of 3D spatial structures.
>
> ---
> [1] WorldPlay: Towards Long-Term Geometric Consistency for Real-Time Interactive World Modeling. Arxiv 2025.
>
> [2] Matrix-game 2.0: An open-source real-time and streaming interactive world model. Arxiv 2025.
>
> [3] Advancing Open-source World Models. Arxiv 2026.
>
> [4] Genie 3: A new frontier for world models. https://deepmind.google/models/genie/ 2025.
>
> [5] Grounded SAM: Assembling Open-World Models for Diverse Visual Tasks. Arxiv 2024.
>
> [6] Depth Anything 3: Recovering the Visual Space from Any Views. ICLR 2026.
>
> [7] Orient Anything V2: Unifying Orientation and Rotation Understanding. NeurIPS 2025.
>
> [8] Laion-aesthetics. https://laion.ai/blog/laion-aesthetics/ 2022.
>
> [9] Pick-a-Pic: An Open Dataset of User Preferences for Text-to-Image Generation. NeurIPS 2023.

---

> > ### Author Rebuttal · Reviewer_eCWH · 2026-04-02
> >
> > My main concern is still about the limited type of evaluation tasks, i.e., only "first-person view." Plus, given the existence of RLVR, the concept of "training WM with RL" is not that novel, making the novelty of this paper kind of weakened. Overall, the novelty and the technical contribution of this work are not good enough, and therefore I'm inclined to maintain the weak reject.

---

> > > ### Author Response · Authors · 2026-04-06
> > >
> > > We appreciate the reviewer’s feedback. We believe it is necessary to further clarify our approach and address the concerns raised.
> > >
> > > ---
> > > ## Concerns on Limited Evaluation Tasks:
> > >
> > > **Support for Diverse Tasks**: As supported in WorldPlay, our method also supports third-person view movement. Furthermore, our current evaluation suite already encompasses third-person interaction cases.
> > >
> > > **Scope of Our Work**: Current limitations regarding interaction types stem from the inherent constraints of existing video diffusion-based world models, which primarily focus on camera movement. Our work aims to provide a post-training framework for these models; extending base model to new forms of interaction inputs is beyond the scope of an post-training framework and belongs to the realm of pre-training.
> > >
> > > **Generalizability**: Once the base model supports new forms of interaction inputs, adapting our framework to these new types depends solely on the design of the reward function (i.e., verifying if the interaction is correctly executed). As a general RL framework, our approach is natively compatible with diverse and evolving reward mechanisms.
> > >
> > > ---
> > > ## Concerns Regarding Differences with RLVR-World:
> > >
> > > We respectfully disagree that RLVR-World diminishes the novelty of our work, as the two studies target fundamentally different "world model" paradigms:
> > >
> > > RLVR-World operates on an LLM-style architecture that autoregressively generates discrete tokens. In contrast, our framework is specifically designed for autoregressive video diffusion-based world model.
> > >
> > > Since RLVR-World remains within the discrete token domain, it can natively inherit the RL strategies used in LLMs. However, applying RL to diffusion-based models introduces unique complexities. Specifically, the time-intensive diffusion process and continuous latent outputs require a complete redesign of the rollout process and fine-grained reward mechanisms—challenges that are non-existent in LLM-like methods.
> > >
> > > Given that the underlying base models and optimization landscapes are entirely different, we believe our technical solutions for video diffusion-based world models represent a distinct and significant contribution to the field.

---

### Official Review · Reviewer_5kDU · 2026-03-03

**Soundness:** 2
**Presentation:** 3
**Significance:** 3
**Originality:** 2
**Overall Recommendation:** 4
**Confidence:** 2

**Summary:**

This paper proposes WorldCompass, which focuses on enhancing the capabilities of long-horizon, autoregressive video generation models (specifically WorldPlay) by employing a post-training reinforcement learning approach. The framework introduces a clip-level rollout strategy, complementary reward functions, and efficient RL optimization to steer the model's exploration towards greater interaction accuracy and visual quality. The authors evaluate the effectiveness of WorldCompass by performing post-training on WorldPlay, reporting substantial improvements in both interaction-following and visual fidelity across various video lengths and action complexities.

**Compliance With Llm Reviewing Policy:**

Affirmed.

**Final Justification:**

The author's has address my main concern, I maintain my score.

**Key Questions For Authors:**

1. Can you provide additional insights into how to mitigate reward hacking? Are there specific environments or tasks where this remains a significant challenge?


2. Could you elaborate on the theoretical basis for using clip-level rollout and complementary reward functions? How do these innovations compare to existing methods, such as those used in sequential or multi-step RL tasks?


3. How does WorldCompass perform with action sequences that significantly diverge from the training data, particularly in extreme or novel environments?

**Limitations:**

Yes

**Strengths And Weaknesses:**

**Strengths:**
1. Comprehensive Evaluation: The authors provide detailed quantitative results (e.g., action-following accuracy and HPSv3 scores) across multiple model variants, scenarios, and video lengths, showcasing the framework’s ability to handle complex, compositional action sequences.


2. Efficiency Optimization: The framework includes strategies like clip-level rollout and Best-of-N sample selection, significantly improving training efficiency without sacrificing performance. The approach is computationally efficient, addressing concerns with traditional RL setups in long-horizon tasks.


3. Timely Problem Domain: The proposed framework leverages reinforcement learning to improve post-training for long-horizon world models, this innovation is a step toward more interactive, consistent video generation in complex environments.

**Weaknesses:**


1. WorldCompass explicitly removes the KL divergence penalty from the DiffusionNFT objective, relying instead on a lower learning rate and the EMA update strategy to prevent over-optimization. This seems mathematically unjustifiable in modern RL, as the KL penalty is exactly what anchors the policy to the prior to prevent catastrophic mode collapse.


2. The paper heavily relies on existing techniques. It takes WorldPlay, changes DiffusionNFT, uses an off-the-shelf 3D estimator for one reward, and HPSv3 for another.


3. Weak and Noisy Evaluation: The evaluation depends entirely on the 3D foundation model to estimate camera trajectories as the "ground truth" for the Interaction Following score.

---

> ### Author Rebuttal · Authors · 2026-03-31
>
> ---
> **W1 Removing the KL regularization:**
>
> Please refer to the response to Reviewer 2cP4's Q2.
>
> ---
> **W2 Heavily relies on existing techniques:**
>
> Our primary objective is to provide a practical solution to the unique challenges of RL-based world models. We clarify that our technical contribution lies not in a new RL algorithm, but in the first systematic RL framework specifically architected for the long-horizon, autoregressive, and interactive nature of video-based world models.
>
> We highlight a similar research paradigm in multi-step RL for LLMs [1,2]. while it utilize established RL algorithms and reward models, its core innovation resides in the structural redesign (transitioning from rewarding a single complete sequence to fine-grained, segment-wise reinforcement).
>
> In our problem, adapting RL to this new "multi-shot" world modeling paradigm presents unique non-trivial challenges in reward function and credit assignment. We believe our framework, by addressing these hurdles, provides significant and timely insights for the community.
>
> ---
> **W3 Evaluation with 3D foundation model:**
>
> We highlight that the generalization capability of 3D foundation models for camera trajectory prediction is widely recognized in the field. To further validate this in our context, we manually annotated camera motions for 200 video clips across diverse indoor and outdoor scenes. Our evaluation shows that combining the 3D foundation model with our multi-threshold strategy achieves an accuracy of 93%, demonstrating the reliability of our automated action evaluation pipeline.
>
> ---
> **Q1 How to mitigate reward hacking:**
>
> We contend that reward hacking primarily stems from noisy or incomplete reward functions. In domains where reward signals are verifiable and comprehensive, such as mathematical problem-solving in LLM RL, reward hacking is significantly mitigated.
>
> To provide a more reliable reward signal for world models, we designed two complementary reward functions that provide mutual constraints, leading to more stable training outcomes. In contrast, using a single reward function often leads to reward hacking. For example, prioritizing only action-following can degrade visual quality, while focusing exclusively on visual quality may result in static video generation.
>
> Furthermore, we believe that incorporating more reward functions that specifically evaluate spatial consistency and autoregressive error accumulation would further constrain the learning process. Since existing reward models do not yet support these capabilities, developing more comprehensive reward functions remains a promising direction for future research.
>
> ---
> **Q2 Innovations of clip-level rollout and complementary reward:**
>
> Regrading clip-level rollout, unlike multi-step LLM RL [1,2], which typically requires a full sequence to reach a verifiable answer, autoregressive video generation allows for independent evaluation of each clip. We exploit this "independent evaluability" by reusing a shared prefix and performing rollouts only at the target position. This significantly reduces redundant computation compared to standard sequence-level rollouts used in text-based reasoning.
>
> Regrading complementary reward, developing a verifiable reward function for visual content is notoriously difficult compared to text. Visual scoring is often noisy and unstable. We employ multiple reward perspectives to act as mutual regularization, ensuring a more stable training signal. Our multi-reward aggregation strategy aligns with the recent GDPO [3] framework (concurrent with this submission) that individual reward normalization before aggregation, which effectively balances different reward scales and prevents any single objective from dominating the gradient.
>
> ---
> **Q3 Generalization to novel action and environments:**
>
> To evaluate the generalization of our method, we use more complex inputs of 4–5 simultaneous actions to generate mid-term videos. Even in these "zero-shot" conditions never seen during training, WorldCompass post-training boost action-following from 16.64% to 47.95% and HPSv3 score from -0.37 to 0.41. This significant leap underscores RL’s power in enabling models to generalize to unseen, high-complexity environments.
>
> ---
> [1] Synthetic Data Generation & Multi-Step RL for Reasoning & Tool Use. COLM 2025.
>
> [2] Tree Search for LLM Agent Reinforcement Learning. ICLR 2026.
>
> [3] GDPO: Group reward-Decoupled Normalization Policy Optimization for Multi-reward RL Optimization. Arxiv 2026.

---

> > ### Author Rebuttal · Reviewer_5kDU · 2026-04-03
> >
> > I would like to thank the authors for their detailed rebuttal and the effort put into the new evaluations, including the manual annotation of the 200 video clips. While the paper tackles an interesting and timely problem, and the zero-shot generalization results (Q3) are positive, the core of the work remains an engineering integration of existing RL methods with heuristic reward balancing. Because the fundamental technical contributions are limited, my initial concerns are not fully resolved. Therefore, I will maintain my current score of 4.

---

### Official Review · Reviewer_v7uX · 2026-03-12

**Soundness:** 3
**Presentation:** 3
**Significance:** 3
**Originality:** 3
**Overall Recommendation:** 4
**Confidence:** 3

**Summary:**

This paper proposes WorldCompass, an RL post-training framework for interactive video-based world models. The key idea are: (1) clip-level rollout of instead of scoring entire video sequences, generate multiple candidates at a single clip position while sharing the prefix, reducing cost from O(N·G) to O(N+G) and providing finer-grained rewards; (2) complementary reward functions for action-following accuracy (via 3D foundation model camera trajectory estimation) and visual quality (HPSv3), which act as mutual regularizers against reward hacking; (3) DiffusionNFT-based optimization with efficiency tricks (timestep subsampling, Best-of-N selection, progressive clip scheduling). Applied to WorldPlay (HunyuanVideo-1.5 and Wan2.2 variants), the method improves composite action accuracy from ~20% to ~55% and boosts visual quality scores across short/medium/long generation horizons.

**Compliance With Llm Reviewing Policy:**

Affirmed.

**Key Questions For Authors:**

- How sensitive are results to the translation threshold values?
- Any compare against standard fine-tuning on curated high-quality action-following data? This would clarify how much of the gain requires RL specifically vs. just better supervision.
- How would this framework extend to non-camera actions (e.g., object interactions)? The current reward design seems tightly coupled to camera movement estimation.

**Limitations:**

The quality drift limitation is honestly acknowledged but only mitigated by conservative training.

**Strengths And Weaknesses:**

**Strengths:**
- The clip-level rollout idea is a well-motivated and practical idea to use. The efficiency argument is clear, and the fine-grained reward signal is a strong over scoring entire long videos.
- The complementary reward design is validated well: IF-only causes visual collapse, VQ-only produce static content. Simple but important findings providing justifications.
- Strong empirical gains on composite actions (20% → 55%) represent a qualitative shift from "failing to follow" to "actually executing". Results are consistent across model backbones and video lengths.
- The ablation study is thorough and well-organized, all cleanly tested.

**Weaknesses:**
- The action space is limited to 8 discrete camera movements. It's unclear how the framework would handle richer interactions like object manipulation, physics, or non-camera actions. The paper doesn't discuss this limitation.
- The action-following reward relies on a 3D foundation model to estimate camera trajectories, then matches against discrete actions using hand-tuned thresholds. This could be fragile, the multiple translation thresholds (0.01–0.05) are a workaround for scale ambiguity, not a solution. How sensitive are results to these choices?
- No comparison against other world models or other post-training approaches beyond the base WorldPlay. The paper only compares "before vs after" RL on the same model. Even comparing against simple SFT on curated high-quality trajectories would help contextualize the RL gains.

---

> ### Author Rebuttal · Authors · 2026-03-31
>
> ---
> **W1&Q3 How to handle richer interactions:**
>
> We clarify that current limitations regarding interaction forms are primarily constrained by the underlying base world model. Existing solutions such as WorldPlay [1], Matrix-Game 2.0 [2], and LingBot-World [3], as well as the cases demonstrated in Google Genie 3 [4], are largely restricted to to first-person or third-person camera movements. Since the base model lacks the inherent capacity for richer interactions like object manipulation, it is not feasible to further enhance these specific skills during the RL post-training stage.
>
> Moreover, when a base model supports new control capabilities, we can strengthen the interaction-following performance by expanding the reward functions. For instance, to support object manipulation, we could integrate object detection [5], depth estimation [6], and orientation estimation [7] to evaluate the alignment between the generated object pose and the input manipulation condition.
>
> ---
> **W2&Q1 Thresholds for action-following reward:**
>
> To demonstrate the reliability and robustness of our multi-threshold strategy, we manually annotated camera motions for 200 video clips across diverse indoor and outdoor scenes. We then evaluated the prediction accuracy of individual thresholds ranging from 0.01 to 0.05, comparing these results against our combined multi-threshold approach.
> |                  | threshold           | Acc      |
> |------------------|---------------------|--------  |
> | single-threshold | 0.01                |    80%    |
> | single-threshold | 0.02                |    73%    |
> | single-threshold | 0.03                |    71%    |
> | single-threshold | 0.04                |    75%    |
> | single-threshold | 0.05                |    76%    |
> | multi-thresholds | [0.01–0.05]         |    93%    |
>
>
> ---
> **W3&Q2 Comparison with more possible method:**
>
> We provide a performance comparison between WorldCompass, DanceGRPO [8], and a baseline utilizing high-quality SFT data. For the SFT baseline, the high-quality data with complex trajectories are filtered from WorldPlay's[1] training data.
>
> |                    | Combined |  | Basic | |
> |--------------------|-------------|----------------|-----------|-------------|
> |                    | Acc | HPSv3 | Acc | HPSv3 |
> | Base Model         |    19.70 | -0.33 | 64.28 | 1.90       |
> | + High-quality SFT   |     26.47 | -0.12 |  66.89  |  2.11   |
> | + DanceGRPO          |     20.02 | 0.59 | 67.43 | 3.97        |
> | + WorldCompass       |     54.82 | 0.73 | 76.56 | 3.72        |
>
> ---
> [1] WorldPlay: Towards Long-Term Geometric Consistency for Real-Time Interactive World Modeling. Arxiv 2025.
>
> [2] Matrix-game 2.0: An open-source real-time and streaming interactive world model. Arxiv 2025.
>
> [3] Advancing Open-source World Models. Arxiv 2026.
>
> [4] Genie 3: A new frontier for world models. https://deepmind.google/models/genie/ 2025.
>
> [5] Grounded SAM: Assembling Open-World Models for Diverse Visual Tasks. Arxiv 2024.
>
> [6] Depth Anything 3: Recovering the Visual Space from Any Views. ICLR 2026.
>
> [7] Orient Anything V2: Unifying Orientation and Rotation Understanding. NeurIPS 2025.
>
> [8] DanceGRPO: Unleashing GRPO on Visual Generation. Arxiv 2025.

---

> > ### Author Rebuttal · Reviewer_v7uX · 2026-04-03
> >
> > Thanks to the authors for clarifying. I would like to keep my current positive score.

---

### Official Review · Reviewer_2cP4 · 2026-03-13

**Soundness:** 3
**Presentation:** 4
**Significance:** 4
**Originality:** 2
**Overall Recommendation:** 4
**Confidence:** 3

**Summary:**

This paper proposes WorldCompass, an RL post-training framework for long-horizon video world models. The method has three main components: clip-level rollout for finer-grained and more efficient training, complementary reward functions for action-following and visual quality, and an efficient optimization recipe based on negative-aware fine-tuning. The method is evaluated on two WorldPlay backbones and reports substantial gains in both action-following accuracy and HPSv3 across various settings, especially on composite actions.

**Compliance With Llm Reviewing Policy:**

Affirmed.

**Final Justification:**

The author's has address my main concern, showing the aesthetics performance doesn't drop.

**Key Questions For Authors:**

## Question 1:
How do you choose the \(\lambda\) values in Eq. (5)? The paper states that they are set to \(2/3\) and \(3\), but it is unclear whether these choices follow any empirical criterion. Could the authors clarify how these values were selected and whether the method is sensitive to them?

## Question 2:
Why is the KL divergence loss used in the original DiffusionNFT omitted here? Removing the KL regularization may make the training more prone to reward hacking, which also relates to my concern in Weakness 2. Could the authors elaborate it?

**Limitations:**

yes

**Strengths And Weaknesses:**

## Strength 1: Practical and valuable settings.
The paper studies an important and underexplored setting: post-training for long-horizon interactive control in video world models. The motivation is clear and practically relevant.

## Strength 2: Good performance.
The method shows large improvements, especially on composite actions, while also improving visual quality. The gains are consistent across two backbones.

## Strength 3: Interesting ablation studies.
The ablation studies are informative and help justify key design choices, especially clip-level rollout and dual rewards.

## Strgenth 4: Various backbones
The method is evaluated on multiple backbones, including Wan2.2 and HY-Video-1.5, which strengthens the empirical results.

## Weakness 1: Limited novelty
The method appears to be more of a careful adaptation and integration of existing group-based RL techniques for diffusion and video models, rather than a fundamentally new RL algorithm. As a result, the technical contribution feels somewhat incremental.

## Weakness 2: Concern of Self-fulifing.
Action-following is evaluated via a 3D foundation model that estimates camera trajectories and then thresholds them into discrete actions; visual quality is measured by HPSv3. This means the entire optimization target is mediated by two external proxies. It is not fully convincing that improvements reflect broader “world-model” capability rather than better optimization toward these proxies. The paper discusses reward hacking, but the validation is limited. It would strengthen the paper to include additional evaluation metrics, such as LLM-based judges or LION Aesthetic score.

## Weakness 3: Lack of details of the training set
The paper would benefit from a more detailed description of the 4k-image training set.

---

> ### Author Rebuttal · Authors · 2026-03-31
>
> ---
> **W1 Limited Novelty:**
>
> Our primary objective is to provide a practical solution to the unique challenges of reinforcement learning for world models. We clarify that our technical contribution lies not in a new RL algorithm, but in the first systematic RL framework specifically architected for the long-horizon, autoregressive, and interactive nature of video-based world models.
>
> We highlight a similar research paradigm in multi-step RL for LLMs [1]. While it utilize established RL algorithms and reward models, its core innovation resides in the structural redesign (transitioning from rewarding a single complete sequence to fine-grained, segment-wise reinforcement).
>
> In our problem, adapting RL to this new "multi-shot" world modeling paradigm presents unique non-trivial challenges in rollout efficiency, reward function and credit assignment. We believe our framework, by addressing these hurdles, provides significant and timely insights for the community.
>
> ---
> **W2 Concern of Self-fulifing:**
>
> We provide additional evaluation results across a broader range of metrics, including LAION Aesthetic Score [2] and PickScore [3]:
>
> | Model           | Aesthetic | PickScore  |
> |-----------------|-----------|------------|
> | HY-Video-1.5    | 5.0762    | 19.2070    |
> | +WorldCompass   | 5.4344    | 19.3831    |
> | Δ               | +0.3583   | +0.1761    |
>
> ---
> **W3 Details of the training set:**
>
> Our 4K-image training set consists of high-quality image-text pairs originally curated for text-to-image generation tasks. The dataset primarily features diverse indoor and outdoor landscapes, with high-quality captions generated by Qwen3-VL [4].
>
> ---
> **Q1 $\lambda$ values in Eq. (5):**
>
> We set $\lambda = 2/3$ to establish a 2:1 ratio between Interaction Following and Visual Quality. This choice aligns with our primary objective: prioritizing the model's ability to follow complex interactive condition, which we think is a more challenging and critical aspect of world modeling.
>
> Our method is not highly sensitive to this hyper-parameter; stable performance improvement can be achieved across various ratios (from 1:1 to 1:3). The value of $\lambda$ primarily affects the performance trade-off between the two objectives. A higher weight in one dimension leads to more performance gains in that area at the slight expense of the other, without affecting the overall stability of the RL training.
>
> ---
> **Q2 Removing the KL regularization:**
>
> Actually, removing KL regularization is a recognized practice when the target RL task requires a significant shift in the model distribution. It has been discussed in recent works such as DanceGRPO [5] for diffusion models and DAPO [6] for large-scale LLM reinforcement learning.
>
> In our specific scenario, the initial model struggles to process complex interactive actions. We intentionally omitted the KL constraint to allow the model sufficient flexibility to adapt to new motion dynamics. Our empirical results confirm that this design choice leads to substantial improvements in action-following performance without compromising training stability.
>
> ---
> [1] Synthetic Data Generation & Multi-Step RL for Reasoning & Tool Use. COLM 2025.
>
> [2] Laion-aesthetics. https://laion.ai/blog/laion-aesthetics/ 2022
>
> [3] Pick-a-Pic: An Open Dataset of User Preferences for Text-to-Image Generation. NeurIPS 2023.
>
> [4] Qwen3-VL Technical Report. Arxiv 2025.
>
> [5] DanceGRPO: Unleashing GRPO on Visual Generation. Arxiv 2025.
>
> [6] DAPO: An Open-Source LLM Reinforcement Learning System at Scale. NeurIPS 2025.

---

> > ### Author Rebuttal · Reviewer_2cP4 · 2026-04-04
> >
> > Thanks for the author's valuable response.
> > My concern has been resolved. I raised my score from 3 to 4.

---

### Decision · Program_Chairs · 2026-04-30

**Decision:**

Accept (regular)

**Comment:**

Reviewers agree that the paper studies an important problem and proposes a simple yet effective idea, with good empirical performance supported by detailed ablations. The main concern was about the limited novelty, as the proposed method is essentially a combination of existing techniques. I think this was nicely addressed by the authors' rebuttal, which argues that the core contribution is not a new RL method but rather "the first systematic RL framework specifically architected for [...] video-based world models". As such I am recommending acceptance and encourage the authors to revise the manuscript following the reviewers' feedback.